# Effect of Inter-Reader Variability on Diffusion-Weighted MRI Apparent Diffusion Coefficient Measurements and Prediction of Pathologic Complete Response for Breast Cancer

**Nu N. Le** [1], **Wen Li** [1,*], **Natsuko Onishi** [1], **David C. Newitt** [1], **Jessica E. Gibbs** [1], **Lisa J. Wilmes** [1], **John Kornak** [2], **Savannah C. Partridge** [3], **Barbara LeStage** [4], **Elissa R. Price** [1], **Bonnie N. Joe** [1], **Laura J. Esserman** [5] and **Nola M. Hylton** [1]

[1] Department of Radiology & Biomedical Imaging, University of California, San Francisco, CA 94158, USA; nu.le@ucsf.edu (N.N.L.); natsuko.onishi@ucsf.edu (N.O.); david.newitt@ucsf.edu (D.C.N.); jessica.gibbs@ucsf.edu (J.E.G.); lisa.wilmes@ucsf.edu (L.J.W.); elissa.price@ucsf.edu (E.R.P.); bonnie.joe@ucsf.edu (B.N.J.); nola.hylton@ucsf.edu (N.M.H.)

[2] Department of Epidemiology and Biostatistics, University of California, San Francisco, CA 94158, USA; john.kornak@ucsf.edu

[3] Department of Radiology, University of Washington, Seattle, WA 98195, USA; spartrid@seattlecca.org

[4] I-SPY 2 Advocacy Group, San Francisco, CA 94143, USA; blestage51@gmail.com

[5] Department of Surgery and Radiology, University of California, San Francisco, CA 94143, USA; laura.esserman@ucsf.edu

[*] Correspondence: wen.li@ucsf.edu

**Abstract:** This study evaluated the inter-reader agreement of tumor apparent diffusion coefficient (ADC) measurements performed on breast diffusion-weighted imaging (DWI) for assessing treatment response in a multi-center clinical trial of neoadjuvant chemotherapy (NAC) for breast cancer. DWIs from 103 breast cancer patients (mean age: $46 \pm 11$ years) acquired at baseline and after 3 weeks of treatment were evaluated independently by two readers. Three types of tumor regions of interests (ROIs) were delineated: multiple-slice restricted, single-slice restricted and single-slice tumor ROIs. Compared to tumor ROIs, restricted ROIs were limited to low ADC areas of enhancing tumor only. We found excellent agreement (intraclass correlation coefficient [ICC] ranged from 0.94 to 0.98) for mean ADC. Higher ICCs were observed in multiple-slice restricted ROIs (range: 0.97 to 0.98) than in other two ROI types (both in the range of 0.94 to 0.98). Among the three ROI types, the highest area under the receiver operating characteristic curves (AUCs) were observed for mean ADC of multiple-slice restricted ROIs (0.65, 95% confidence interval [CI]: 0.52–0.79 and 0.67, 95% CI: 0.53–0.81 for Reader 1 and Reader 2, respectively). In conclusion, mean ADC values of multiple-slice restricted ROI showed excellent agreement and similar predictive performance for pathologic complete response between the two readers.

**Keywords:** reader variability; diffusion-weighted imaging; breast cancer; treatment response; neoadjuvant therapy

## 1. Introduction

Neoadjuvant chemotherapy (NAC) is one of the standard treatments for early, high-risk breast cancer. NAC provides an opportunity to use imaging or biological markers to monitor tumor response at various time points during treatment. Diffusion-weighted imaging (DWI) is a non-contrast MR imaging technique based upon measuring the random motion of water molecules within tissue. The apparent diffusion coefficient (ADC) is a quantitative measure derived from at least two DWI images with different b-values. Many breast DWI studies have shown that tumor ADC can provide valuable information for evaluating tumor response in the NAC setting [1–4], and can provide distinct information from

quantitative measurements provided by dynamic contrast-enhanced (DCE) MRI. Compared to DCE-MRI, breast DWI often has poorer image quality due to artifacts and lower spatial resolution, and lacks standardization in image acquisition, image interpretation, and ADC calculation [5], factors that limit its widespread use in clinical practice.

Recently, a large multi-center clinical trial, the American College of Radiology Imaging Network (ACRIN) 6698 trial [6,7], conducted as a sub-study of the Investigation of Serial Studies to Predict Your Therapeutic Response through Imaging and Molecular Analysis 2 (I-SPY 2 TRIAL) demonstrated that change in mean ADC after 12 weeks of therapy (inter-regimen) was predictive of pathologic complete response (pCR) [6]. A test–retest study using a sub-cohort of ACRIN 6698 patient scans reported excellent repeatability and reproducibility of tumor mean ADC measurements, with intraclass correlation coefficient (ICC) of 0.92 [95% CI: 0.80–0.97] [8]. However, it was evaluated only in a small cohort ($n$ = 20).

Various ROI delineation methods and inter-reader studies of ADC measurements have been reported [9–14]. Most focused on the comparison of mean ADC values [10–12], and several studied the impact of ROI placement methods on the diagnostic performance of ADC [13–15]. Very few studies analyzed the impact of ROI placement methods on the evaluation of response to NAC for breast cancer [9]. Van Heeswijk et al. included histogram metrics (min, max, median, 5th to 95th percentiles) in addition to mean ADC in their study of rectal tumors. All these studies above used imaging data acquired at a single institution. In this paper, we describe a retrospective analysis of an inter-reader study of three types of ROIs on the prediction of pathologic complete response (pCR) using data from I-SPY 2, a multi-center NAC clinical trial.

## 2. Materials and Methods

### 2.1. Patient Population

Women who are 18 years of age or older, are diagnosed with clinical stage II or III breast cancer, and never had previous treatment of surgery or systemic therapy for this cancer, are eligible to participate in the I-SPY 2 TRIAL. The tumor size should be at least 2.5 cm measured by clinical assessment or by imaging. Hormone receptor (HR)-positive and human epidermal growth factor receptor 2 (HER2)-negative cancer with low risk assessed by MammaPrint (Agendia, Amsterdam, The Netherlands) are excluded from the trial.

A cohort of 249 women enrolled in I-SPY 2 and randomized to pembrolizumab plus standard or corresponding control arm (standard chemotherapy) at qualified study centers were considered for inclusion in this analysis. The I-SPY 2 TRIAL (ClinicalTrials.gov identification number: NCT01042379) is HIPAA-compliant and was performed under Institutional Review Board (IRB) approval. All patients gave informed consent prior to enrolling and before starting treatment. All patients had human epidermal growth receptor 2 negative breast cancer, verified at baseline. The primary endpoint of I-SPY 2 is pCR, defined as the absence of invasive tumor in breast and lymph nodes at the time of surgery.

### 2.2. Imaging Acquisition

The MRI component of the I-SPY 2 trial consisted of four sequential MRI exams acquired: before NAC (T0), after 3 weeks of NAC (T1), inter-regimen (T2), and pre-surgery (T3). In this study, we used MRI examinations at T0 and T1. MRI examinations were performed on 1.5 T or 3 T scanners across a variety of vendor platforms and institutions using a dedicated breast coil and prospectively defined protocol. DWI-MRI was performed using a fat-suppressed single-shot echo planar imaging sequence with the parameters TR $\geq$ 4000 ms, TE = 50–100 ms, FOV = 260–360 mm to achieve full bilateral coverage, acquisition matrix = 128–192 with in-plane resolution $\leq$ 1.9 mm, slice thickness = 3–5 mm, slice gap $\leq$ 1 mm, and number of signal averages $\geq$ 2. Diffusion weighting b-values of 0 and 800 s/mm$^2$ were specified, with an acquisition time no longer than 5 min. DCE-MRI was also acquired during the same MRI scanning. Three-dimensional fat-suppressed T1

were acquired before and after injection of a gadolinium contrast agent. Post-contrast imaging was started simultaneously with injection. Phase duration was 80–100 s with a minimum of 8 min of imaging following injection.

### 2.3. Image Analysis

A standardized quality ranking system was used to evaluate DWI studies for the three image-quality categories: (1) artifacts, (2) fat suppression, and (3) signal-to-noise ratio (SNR). One of the readers (WL) evaluated the study and gave an overall quality rating of poor, moderate, or good. "Poor" image quality refers to severe artifacts and/or failed fat suppression and/or low signal-to-noise ratio in the tumor area in either original DWI at $b = 800$ s/mm$^2$ or derived ADC map. "Moderate" image quality refers to when original DWI or derived ADC map has issues in one or more of the categories above, but ROI delineation is still possible. "Good" image quality refers to no obvious issues in any of the three categories above. Poor-quality exams were considered not analyzable and were excluded from the study. Moderate- and good-quality studies were then evaluated as analyzable or not analyzable based on the degree to which any negative quality issues were found to prevent confident definition of a ROI. Three patients with missing pCR outcome were excluded (see Figure 1 for details on data inclusion/exclusion).

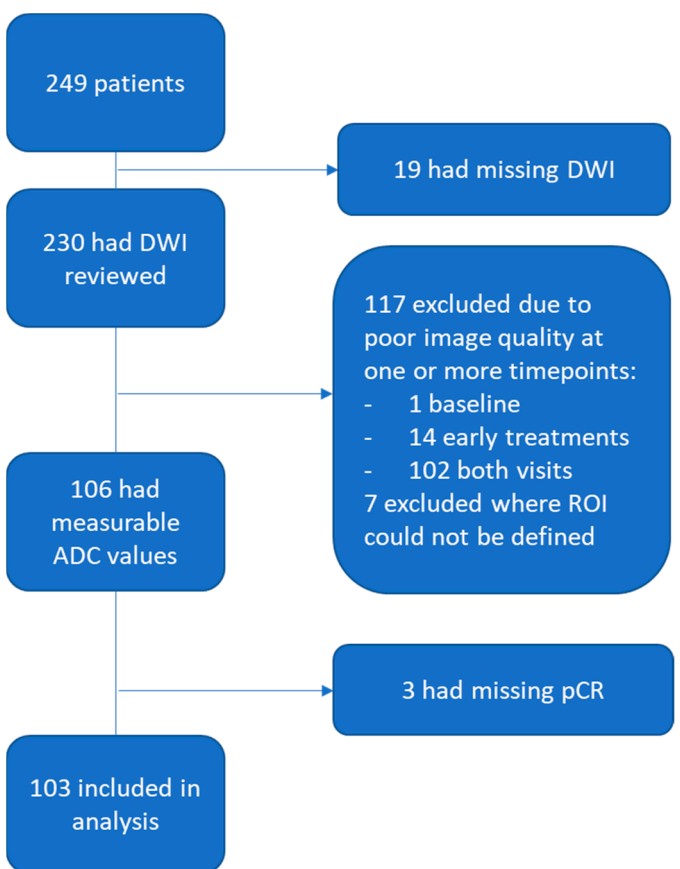

**Figure 1.** Study patient inclusion and exclusion flowchart. From 249 I-SPY 2 patients, 103 were included in this analysis.

ROI evaluation was performed on DWI images acquired at T0 and T1 by two readers with 7 and 0 years of experience in breast DWI, blinded to pathologic outcomes. Each reader was trained on a DWI cohort of 30 patients acquired at T0 and T1 (60 exams in total). The training set was randomly selected from two other drug arms in I-SPY 2 with matching sites and HR positive/negative ratio as the full cohort (*n* = 249) in this study. Disagreement or difficult cases were discussed with a breast radiologist.

DWI, ADC map and DCE subtraction images were simultaneously checked to localize the lesion. The tumor was then localized on the b = 0 DWI image using similar breast anatomical structures to those seen on subtracted DCE images and was delineated on the ADC map to encompass areas with low ADC values and high signal intensity in the b = 800 DWI. Surrounding fat and tissue were excluded to eliminate partial volume effects. Care was taken to avoid cystic or necrotic regions and areas exhibiting T2 shine-through.

Three types of breast tumor ROIs were analyzed in this study: multiple-slice restricted ROI, single-slice restricted ROI, and single-slice tumor ROI. Table 1 describes each ROI delineation technique in detail. Two types of ROI—multiple-slice restricted ROI and single-slice tumor ROI—were manually delineated (see Figure 2 for example cases). The third type of ROI—single-slice restricted ROI—was automatically generated from multiple-slice restricted ROI on the same axial slice as the single-slice tumor ROI. Multiple-slice ROIs were delineated on all slices where tumor could be seen. Restricted ROIs focused on lower ADC areas only while tumor ROIs enclosed the whole tumor area that could be seen in both DWI and DCE-MRI. Special care was taken to place ROIs at the same locations for the same lesion in T0 and T1. The in-house software developed using IDL (Exelis Visual Information Solutions, Boulder, CO) was used to calculate ADC based on the classic mono-exponential decay model: $ADC = [\ln(S\_0) - \ln(S\_800)]/800$, where $S\_0$ and $S\_800$ are signal intensities acquired with b-values of 0 and 800 s/mm$^2$.

**Table 1.** ROI delineation Approach.

| ROI Type | Description |
|---|---|
| Multiple-slice restricted ROI | Manually sampled area with most restricted diffusion (low ADC) defined within tumor (contrast-enhanced on DCE) and covering all axial slices where tumor was seen |
| Single-slice restricted ROI | Single slice selected from the multiple-slice restricted ROI and matching level of single-slice tumor ROI |
| Single-slice tumor ROI | Manually sampled area corresponding to the full tumor seen on DCE-MRI, defined on the single axial slice with largest tumor area |

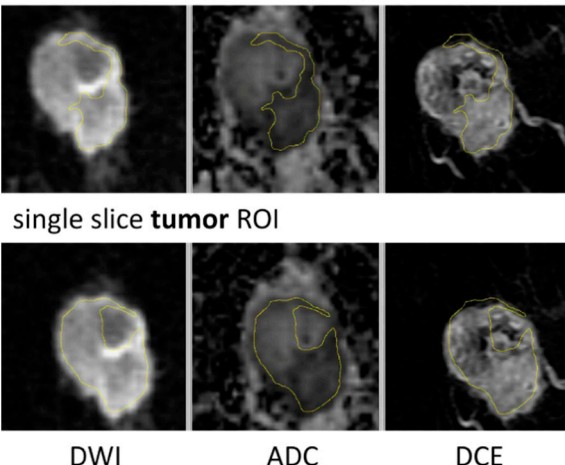

**Figure 2.** ROI delineation examples for single-slice restricted ROI and single-slice tumor ROI.

ADC metrics—mean and percentiles of the ADC histogram (minimum, 5th percentile, 15th percentile, 25th percentile, 50th percentile, 75th percentile, 95th percentile, maximum)—were automatically calculated from all three types of ROIs delineated by each reader at T0

and T1, respectively. Percentage changes in ADC metrics from T0 to T1 were also calculated to evaluate effect of inter-reader variability on treatment response.

### 2.4. Statistical Analysis

The variability of ADC measurements between two readers was evaluated using the ICC [16]. The ICCs were calculated with the irr package version 0.84.1 in R version 4.0.3 (R Foundation for Statistical Computing, Vienna, Austria) [17]. Predictive performances of percentage change in ADC metrics from T0 to T1 were assessed by area under the receiver operating characteristic curves (AUCs) to illustrate the tradeoff between sensitivity and specificity on various cutoff values in the prediction of pCR. *p*-values of difference between two AUCs were determined using the DeLong test. The pROC package in R version 4.0.3 was used to analyze receiver operating characteristic (ROC) curves and calculate AUCs [18].

## 3. Results

### 3.1. Patient Characteristics

A total of 249 patients participating in I-SPY 2 were evaluated for inclusion in this study (original cohort). Of those, 103 patients (analyzed cohort) had DWI data that could be assessed (see Figure 1 for patient exclusion details). Of those 103, 30 had a pCR and 73 had a non-pCR after NAC. See Table 2 for patient characteristics. Overall demographics agreed well between the original and analyzed cohort, with some minor differences observed in age and subtype distributions.

**Table 2.** Patient characteristics.

| Characteristics | Original Cohort (N = 249) | Analyzed Cohort (N = 103) |
|---|---|---|
| Age | 49 ± 11 | 46 ± 11 |
| Race | | |
| - White | 191 (77%) | 76 (74%) |
| - Black or African American | 32 (13%) | 15 (15%) |
| - Asian | 16 (6%) | 8 (8%) |
| - American Indian or Alaska Native | 4 (2%) | 0 (0%) |
| - Others | 5 (2%) | 3 (3%) |
| Cancer Subtype | | |
| - HR+/HER2- | 134 (54%) | 62 (60%) |
| - HR−/HER2- | 114 (46%) | 41 (40%) |
| Treatment | | |
| - Paclitaxel | 180 (72%) | 75 (73%) |
| - Paclitaxel + Pembrolizumab | 69 (28%) | 28 (27%) |
| Menopausal Status | | |
| - Premenopausal | 126 (51%) | 58 (56%) |
| - Postmenopausal | 72 (29%) | 26 (25%) |
| - Perimenopausal | 13 (5%) | 7 (7%) |
| - Others | 38 (15%) | 12 (12%) |
| Pathologic outcome | | |
| - pCR | 64 (26%) | 30 (29%) |
| - non-pCR | 174 (70%) | 73 (71%) |
| - Unknown | 11 (4%) | 0 (0%) |

### 3.2. Inter-Reader Variability

ICCs between two readers for three types of ROIs and nine types of ADC metrics extracted from each ROI type are plotted in Figure 3. All the estimated ICC values and

corresponding 95% confidence intervals in Figure 3 can be found in Supplemental Table S1, and the results are summarized below.

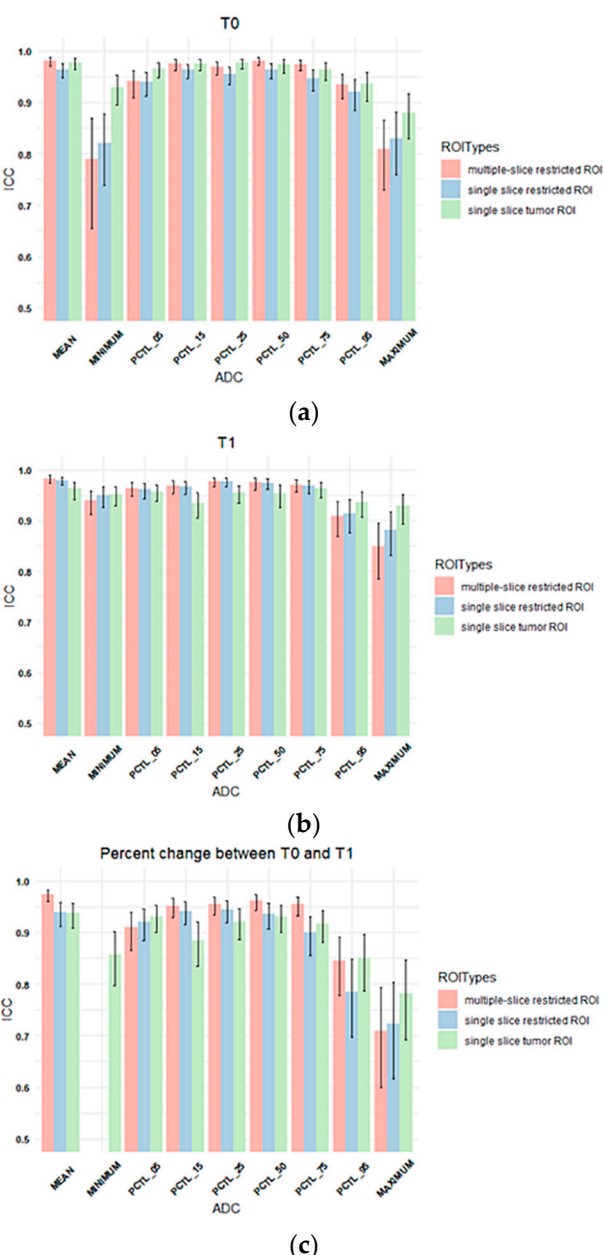

**Figure 3.** Column plots showing the ICC values of tumor ADC histogram parameters between Reader 1 and Reader 2 for (**a**) T0, (**b**) T1, and (**c**) percentage change between T0 and T1. The x-axis lists mean ADC and histogram parameters of ADC values within the ROI (minimum, 5th percentile, 15th percentile, 25th percentile, 50th percentile, 75th percentile, 95th percentile, maximum), and the y-axis shows the ICC values. Colors represent ROI types. Multiple-slice and single-slice restricted ROI were shown as blank for minimum ADC in (**c**) because both ICCs were below 0.5. (pink: multiple-slice restricted ROI, blue: single-slice restricted ROI, green: single-slice tumor ROI). Error bar is 95% CI. PCTL = percentile.

The highest ICCs were observed for mean ADCs (range: 0.96 to 0.98). ADC metrics at T1 generally showed higher estimated ICCs (range: 0.84 to 0.98) compared to those at T0 (range: 0.79 to 0.98). Percentage changes in ADC metrics from T0 to T1 showed lower ICCs compared to absolute ADC values at T0 or T1, ranging from 0.045 to 0.97 with lowest estimated ICCs observed for percentage change in the minimum ADC when restricted ROIs

were used (0.045 and 0.083) for multiple-slice and single-slice restricted ROIs, respectively, Supplemental Table S1).

For percentage changes in mean ADC from T0 to T1, ICCs were similar and high among three types of ROIs although the multiple-slice restricted ROI achieved slightly higher estimated ICC (0.97) than the other two ROI types (both were 0.94). However, higher ICCs were observed when single-slice tumor ROI was used to extract other histogram metrics, in particular percentage change in minimum ADC (ICC = 0.86 versus 0.045 [multiple-slice restricted] or 0.083 [single-slice restricted]).

Bland–Altman plots comparing percentage change in mean tumor ADC measured by two readers for each ROI type are shown in Figure 4. High agreement without systematic bias between two readers was indicated. Multiple-slice restricted ROI showed narrower 95% limits of agreement between readers compared to the other two ROI types for measuring percentage change in tumor mean ADC. Mean tumor ADC measured by two readers for each ROI type is shown in Table 3.

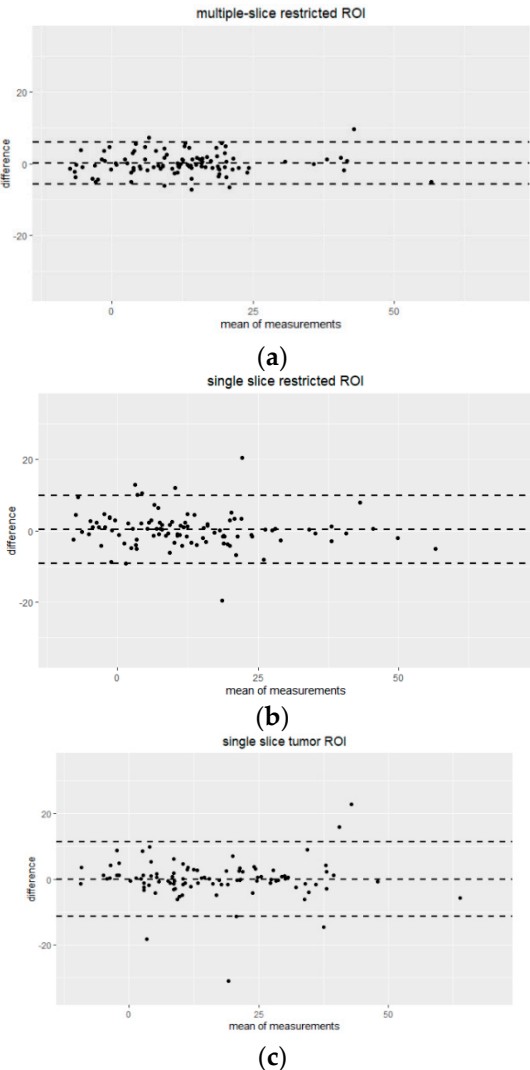

**Figure 4.** Bland-Altman plots showing the agreement of percentage change in mean tumor ADC measurements by Reader 1 and Reader 2 for three different ROI approaches: (**a**) multiple-slice restricted ROI, (**b**) single-slice restricted ROI and (**c**) single-slice tumor ROI. The x-axis shows the mean of measurements (percentage change in mean tumor ADC), and the y-axis shows the difference between measurements from two readers. The unit for both axes are %. The mean difference across all measurements is represented by the horizontal dotted line in the middle and 95% limits of agreement (mean ± 1.96 SD) are shown as horizontal dotted lines above and below the mean.

**Table 3.** Mean tumor ADC measured by two readers for each ROI type (unit: $\times 10^{-3}$ mm$^2$/s).

| Timepoint | Multiple-Slice Restricted ROI | | Single-Slice Restricted ROI | | Single-Slice Tumor ROI | |
|---|---|---|---|---|---|---|
| | Reader 1 | Reader 2 | Reader 1 | Reader 2 | Reader 1 | Reader 2 |
| T0 | $0.930 \pm 0.109$ | $0.935 \pm 0.116$ | $0.926 \pm 0.112$ | $0.932 \pm 0.120$ | $0.973 \pm 0.156$ | $0.984 \pm 0.153$ |
| T1 | $1.031 \pm 0.148$ | $1.037 \pm 0.151$ | $1.030 \pm 0.159$ | $1.033 \pm 0.161$ | $1.098 \pm 0.192$ | $1.114 \pm 0.202$ |

Inter-Reader Variability on Predicting Pathologic Complete Response

The AUC was estimated to evaluate the performance of the percentage change in each tumor ADC metric from T0 to T1 to predict pCR. The comparison of AUCs between two readers for three types of ROIs is shown in Figure 5. The corresponding AUC values are shown in Supplemental Table S2. Overall, AUC values were similar between two readers across all tumor ADC metrics for all three ROI types, which is consistent with reader agreement results (ICCs). The estimated AUCs ranged from 0.49 to 0.67 for multiple-slice restricted ROI, 0.54–0.64 for single-slice restricted ROI and 0.38–0.60 for single-slice tumor ROI. Among all ADC metrics, mean tumor ADC showed highest estimated AUC for both readers when multiple-slice restricted ROI or single-slice restricted ROI were used. When single-slice tumor ROI was used, minimum ADC showed the highest estimated AUC for both readers.

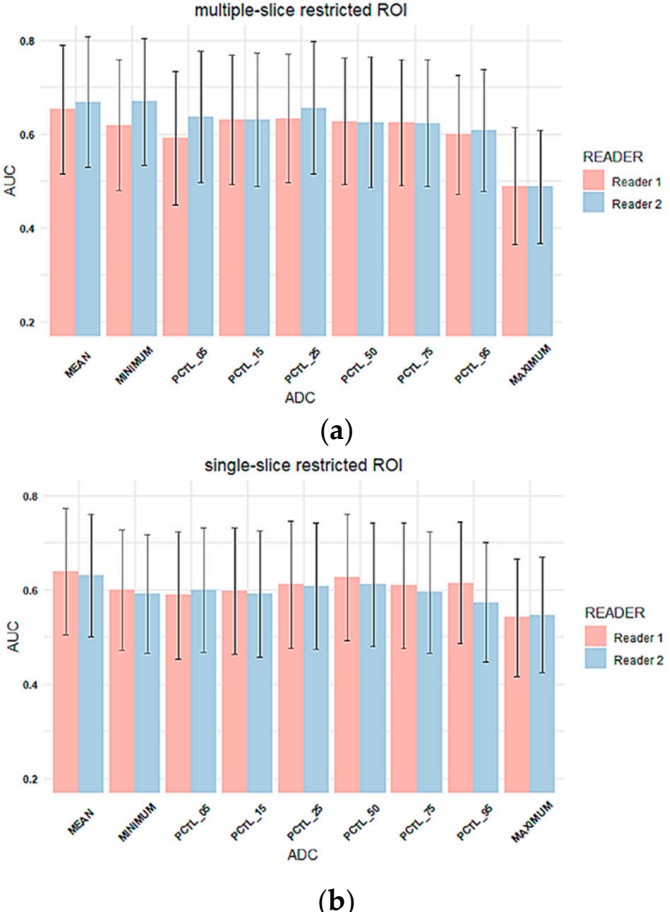

**Figure 5.** *Cont.*

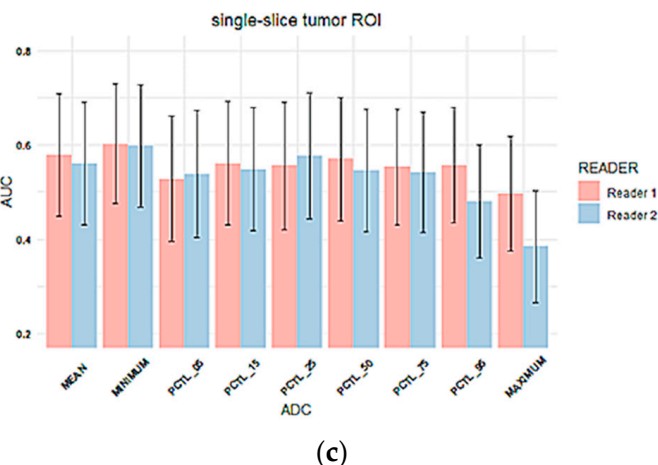

**(c)**

**Figure 5.** Column plots of AUC values for percentage change in tumor ADC metrics for (**a**) multiple-slice restricted ROI, (**b**) single-slice restricted ROI and (**c**) single-slice tumor ROI. In each plot, the x-axis lists mean ADC and histogram parameters of ADC values within the ROI (minimum, 5th percentile, 15th percentile, 25th percentile, 50th percentile, 75th percentile, 95th percentile, maximum), and the y-axis represents estimated AUC values. Colors represent readers. (Pink: reader 1, blue: reader 2). Error bar is 95% CI. PCTL = percentile.

No inter-reader AUC differences were observed for percentage change in mean tumor ADC using any of the ROI methods: Reader 1 versus Reader 2 AUCs were 0.65 (95% CI: 0.52 to 0.79) versus 0.67 (95% CI: 0.53 to 0.81; $p$ = 0.26) for multiple-slice restricted ROI, 0.64 (95% CI: 0.50 to 0.77) versus 0.63 (95% CI: 0.50 to 0.76; $p$ = 0.70) for single-slice restricted ROI, and 0.58 (95% CI: 0.45 to 0.71) versus 0.56 (95% CI: 0.43 to 0.69; $p$ = 0.70) for single-slice tumor ROI, Figure 5. Both ROCs for single-slice tumor ROI were close to the 0.5 diagonal line, suggesting minimal predictive value using this ROI approach (Figure 6). An apparent separation was observed in the mid-range of sensitivity and specificity for both the multiple-slice restricted and single-slice restricted ROIs (between 0.2 and 0.8 for both) with larger gains in sensitivity as specificity increased. Youden index was used to determine the optimal cut points (Table 4). They are at high sensitivity (80% to 93%) and moderate specificity (33% to 57%) for predicting pCR. This indicates that applying different ROI delineation strategies will improve the accuracy of identifying non-pCRs (negatives) when accuracy of identifying pCRs remains high.

**Table 4.** Sensitivity and Specificity at optimal operating point using Youden's Index.

| ROI Types | Reader | Sensitivity | Specificity |
|---|---|---|---|
| Multiple-slice restricted ROI | 1 | 0.82 | 0.57 |
| | 2 | 0.90 | 0.53 |
| Single-slice restricted ROI | 1 | 0.81 | 0.53 |
| | 2 | 0.80 | 0.50 |
| Single-slice tumor ROI | 1 | 0.93 | 0.33 |
| | 2 | 0.89 | 0.33 |

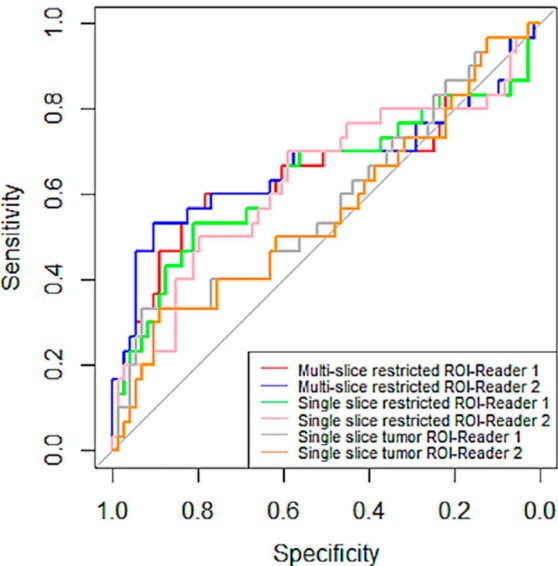

**Figure 6.** ROC curves of percentage change in mean tumor ADC by reader and ROI type. ROI curves for multiple-scheme 1. and blue (Reader 2). The curves in green and pink are for single-slice restricted ROI by Reader 1 and Reader 2, respectively. The last two pairs are curves in gray and tan, representing ROC curves for single-slice tumor ROI by Reader 1 and Reader 2, respectively. The gray straight diagonal line represents AUC = 0.5.

## 4. Discussion

In this multi-center study, we investigated the inter-reader variability of DWI as a marker of breast cancer response to therapy using different tumor ADC metrics and ROI delineation approaches. In general, our results showed similar agreement of tumor ADC metrics extracted from ROIs delineated by two readers compared to other previous breast cancer studies [8,12]. Furthermore, predictive performances of changes in these ADC metrics (after 3 weeks of treatment) generated from different readers were also very close. Overall, these results indicate good reproducibility of quantitative breast tumor ADC measurements using manually delineated ROIs in DWI-MRI.

Manual delineation of ROIs on patient image data inevitably has a subjective component. ICC values reported from our study demonstrated high reproducibility of mean tumor ADC (ICC > 0.96) evaluated either at pretreatment (T0) or at early post-treatment (T1, 3 weeks after treatment initiation) compared to what has been achieved in the previous studies based on other breast tumor ADC metrics [8,19,20]. The ICC value broadly agrees with the ICC reported in the ACRIN 6698 test–retest study [8] where in a small subset of the DWI exams ($n = 20$), the ICC of mean ADC in the whole-tumor ROI between two readers was estimated to be 0.92. However, it was not clear for which treatment time points these 20 exams were performed (T0 or T1). Jang et al. assessed reproducibility of ADC measurements in malignant breast masses [12]. Their manual ROI delineation method was similar to the single-slice tumor ROI in this study. In that study, two radiologists with three and six years of experience interpreted breast MR images independently with an estimated ICC of 0.751 (95% CI: 0.573 to 0.855) in a cohort of 66 patients. The reason a higher ICC was achieved in the present study could be that the two readers were trained using a training set in advance, and consensus was made in the separate training set before readers started the ROI delineation in the cohort ($n = 103$) of this study.

Three types of ROIs were analyzed in our study. The first type was multiple-slice restricted ROI, which was designed to cover the most diffusion-restricted area of the lesion on multiple slices. The second type—single-slice restricted ROI—was automatically extracted from the first using only one axial slice that showed the largest lesion area. The third type was the single-slice tumor ROI, designed to cover the whole lesion on the same axial slice as the second type. The rationale of testing the reproducibility of these three

ROI types was that whole-tumor ROI and restricted ROI are the most commonly used manual delineation methods in the literature [6,14,21]. The use of small ROIs focusing on the most diffusion-restricted lesion area is recommended by the European Society of Breast Radiology (EUSOBI) and supported by published studies [19,21,22]. Our results showed comparable reproducibility of ADC measurements among different types of ROIs, except minimum and maximum ADCs, where tumor ROI achieved higher ICC than restricted ROIs. This observation was more obvious for percentage changes in ADC measurements, which indicates that percentage changes in minimum or maximum tumor ADCs may not be as reliable as other ADC measurements to reflect changes induced by treatment. The ICCs for percentage changes in minimum ADC were 0.045 and 0.083 for multiple-slice and single-slice restricted tumor ROIs, respectively. For comparison, the ICC for percentage change in minimum ADC for single-slice tumor ROI was 0.86. This result suggests that minimum ADC in restricted ROIs may be more subject to inter-reader variability than tumor ROI.

Results from the ACRIN 6698 clinical trial indicate that the percentage change in tumor ADC is predictive of pathologic complete response [6,8]. Therefore, we evaluated the reproducibility of percentage change in ADC measurements in this study. For mean tumor ADC, the highest ICC was found in multiple-slice restricted ROIs, which was confirmed by the Bland–Altman plot in Figure 4. However, it was not observed at the lower (minimum and 5th percentile) and higher spectrums (95th percentile and maximum) of the histogram. ACRIN 6698 also reported ADC histogram metric reproducibility results from a small sample ($n = 20$) [8], in which the highest inter-reader reproducibility was observed at low percentiles (15th and 25th). It is interesting to note those study findings align well with our results if we take into account differences in ROI delineation approach. ACRIN 6698 used a 3D whole-tumor ROI approach, where low percentiles likely represent ADC values of the more restricted part of the tumor, which is comparable to mean ADCs for our multi-slice restricted ROI approach, which demonstrated the highest reproducibility in our study.

This study also evaluated the predictive performances of the percentage change in ADC metrics by ROI type and by reader. Overall, similar AUC values were observed for percentage change in ADCs from Reader 1 and Reader 2. Interestingly, highest AUCs were observed based on mean ADC for restricted ROIs but minimum ADC for single-slice tumor ROI. This observation suggests that the most restricted area of the lesion could be more reflective of treatment response. However, minimum ADC may suffer from poor reproducibility.

This study has several limitations. First, the three types of ROIs could be highly correlated, especially when some ROIs happened to be small in all three types. In addition, the single-slice restricted ROI was completely included in the multiple-slice restricted ROI. This might have led to similar ICC results for multi-slice restricted ROI and single-slice restricted ROI. Second, whole-tumor ROIs were not delineated for a full comparison of restricted versus tumor ROIs. Third, nearly 50% of the original cohort was not analyzable because of poor image quality or other factors preventing ROI delineation, which reduced the sample size by half.

## 5. Conclusions

In conclusion, this multi-center retrospective study found that mean ADCs measured in restricted ROIs were highly reproducible by manual delineation. The mean ADC of multi-slice restricted ROI showed the highest reproducibility. The AUC estimates and confidence intervals were similar in the predictive performance of percentage change in mean ADCs resulting from different readers for all three types of ROIs.

**Supplementary Materials:** The following are available online at https://www.mdpi.com/article/10.3390/tomography8030099/s1, Table S1: ICC values for ADC metrics, Table S2: AUC values for predicting pCR using ROIs delineated by two readers.

**Author Contributions:** Conception and design: W.L., N.M.H. and D.C.N.; Development of statistical methodology: W.L. and J.K.; Data acquisition and interpretation: N.N.L., W.L., N.O., L.J.W., J.E.G., B.N.J., E.R.P. and L.J.E.; Data analysis: N.N.L., W.L. and J.K.; Manuscript writing: N.N.L., W.L., L.J.W., J.E.G., N.O., B.L., S.C.P. and N.M.H. All authors have read and agreed to the published version of the manuscript.

**Funding:** This research was funded by National Cancer Institute of the Institutes of Health grant R01 CA132870, U01 CA225427, P01 CA210961, R01 CA255442, and UCSF Breast Oncology Program: Research Development Program.

**Institutional Review Board Statement:** This study was a retrospective study using data collected from the I-SPY 2 TRIAL. The I-SPY 2 TRIAL was conducted in accordance with the Declaration of Helsinki, and approved by the Institutional Review Board (or Ethics Committee) of each participating institute.

**Informed Consent Statement:** Informed consent was obtained from all subjects enrolled in the I-SPY 2 TRIAL.

**Data Availability Statement:** Imaging and clinical data from this study will be made available upon request in accordance with I-SPY 2 Data and Publication Committee policies.

**Acknowledgments:** The authors would like to thank patients, investigators, site radiologists and coordinators in the I-SPY 2 TRIAL.

**Conflicts of Interest:** The authors declare no conflict of interest.

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
