# Peer review of "Effect of Inter-Reader Variability on Diffusion-Weighted MRI Apparent Diffusion Coefficient Measurements and Prediction of Pathologic Complete Response for Breast Cancer"

_tomography, doi:10.3390/tomography8030099_

Round 1
Reviewer 1 Report
Authors have explored the inter-reader variability of DWI as a marker of breast cancer response to NAC using different tumor ADC metrics. Results were clearly presented and discussed.
Minor comments:
1. Section number for 'Image analysis' is missing.
2. Line 90, authors should consider providing the details of the standardized quality ranking system used for classifying the image quality into poor, moderate, or good for the sake of completeness.
3. Please align the subplots of Figure 3 with fixed indentation.
Author Response
Thank you for your kind words and your time to review our manuscript. We revised it according to your comments. Please see details below:
- Section number for 'Image analysis' is missing
Response: Thank you for pointing it out. We fixed the section number.
- Line 90, authors should consider providing the details of the standardized quality ranking system used for classifying the image quality into poor, moderate, or good for the sake of completeness
Response: Details were added. Please see line 110 to 115 in the revised version.
- Please align the subplots of Figure 3 with fixed indentation.
Response: subplots are aligned now.
Reviewer 2 Report
Please address my comments listed in the attached file

Author Response
We are glad that you found our research interesting. We would like to thank you for spending time to read the manuscript carefully and writing detailed review comments. Please see our responses below:
- Page 1. The authors should clarify three abbreviations in the Abstract: AUC in line 25, CI in line 26, pCR in line 28. Independently from the Abstract, please define abbreviations: MR in line 36 of page 1, ROI in line 54 of page 2. The abbreviations should be explained in the Abstract and in the text of the manuscript independently because they could be published and read separately.
Response: Thank you for noticing our mistakes. We spelled out all abbreviations in the Abstract.
- Page 2, line 69: Please provide references for two statements: “HIPAA-compliant” and “Institutional Review Board (IRB) approval”. Only with the available references, the presented results could be checked and reproduced.
Response: Thank you for your comment. We added a clinical trial identification number from which readers can find all detailed HIPPA and IRB documents.
- In Section 2, the clinical information for the examined breast cancer is not provided. Information about the tumor type, its grade and size, mammographic breast density, and lesion morphology from on DCE-MRI is absent in this paper. Therefore, the dependence of presented materials to clinical conditions could not be evaluated. It seems as a flaw of this study.
Response: Thank you for your comments. Yes. Clinical information about the disease was missing in Section 2. It’s our oversight. We apologize for it. We added the information in the revised version. Please see line 73 to 78.
- Pages 7-8: The units of the values presented in Figure 4 should be presented in both the axis: horizontal [mean ADC x 10-3 mm2/s] and for the vertical axis [delta ADC, %]. Without units, the use of these results could be doubtful. In Figures 3 and 5 the metrics of ADC should be defined.
Response: Thank you for your comments. We added ADC metrics definition in Figures 3 and 5. However, in Figure 4, both axes should have unit of %. We used Bland-Altman to show agreement of percent change in ADC measured by Reader 1 and Reader 2. In this case, the x-axis is the measurement which is the percent change in ADC and the y-axis is the difference in the measurement. We hope our explanation can clarify our misunderstandings.